# Systematic review and meta-analysis comparing Adjustable Transobturator Male System (ATOMS) and Adjustable Continence Therapy (ProACT) for male stress incontinence

**Javier C. Angulo**[1,2☯]*, **Sandra Schönburg**[3], **Alessandro Giammò**[4], **Francisco J. Abellán**[1], **Ignacio Arance**[1,2], **David Lora**[5,6,7☯]

**1** Departamento Clínico, Universidad Europea de Madrid, Madrid, **2** Servicio de Urología, Hospital Universitario de Getafe, Madrid, Spain, **3** Department of Urology and Kidney Transplantation, Martin Luther University, Halle (Saale), Germany, **4** Department of Neuro-Urology, CTO-Spinal Unit, Città della Salute e della Scienza di Torino, Turin, Italy, **5** Instituto de Investigación Sanitaria Hospital "12 de Octubre" (i+12), Madrid, Spain, **6** CIBER de Epidemiología y Salud Pública (CIBERESP), Madrid, Spain, **7** Universidad Complutense de Madrid, Madrid, Spain

☯ These authors contributed equally to this work.
* javier.angulo@salud.madrid.org

**Data Availability Statement:** All relevant data are within the paper and its Supporting Information files.

## Abstract

### Background and purpose

Urinary incontinence is one of the most serious complications of prostate cancer treatment. The objective of this study was to assess efficacy and safety of Adjustable Transobturator Male System (ATOMS) compared to Adjustable Continence Therapy (proACT) for male stress urinary incotinence according to literature findings.

### Material and methods

A systematic review and meta-analysis on adjustable devices ATOMS and ProACT is presented. Studies on female or neurogenic incontinence were excluded. Differences between ATOMS and proACT in primary objective: dryness status (no-pad or one safety pad/day) after initial device adjustment, and in secondary objectives: improvement, satisfaction, complications and device durability, were estimated using random-effect model. Statistical heterogeneity among studies included in the meta-analysis was assessed using tau2, Higgins´s I2 statistics and Cochran´s Q test.

### Results

Combined data of 41 observational studies with 3059 patients showed higher dryness (68 vs. 55%; p = .01) and improvement (91 vs. 80%; p = .007) rate for ATOMS than ProACT. Mean pad-count (-4 vs. -2.5 pads/day; p = .005) and pad-test decrease (-425.7 vs. -211.4 cc; p < .0001) were also significantly lower. Satisfaction was higher for ATOMS (87 vs. 56%; p = .002) and explant rate was higher for proACT (5 vs. 24%; p < .0001). Complication rate for ProACT was also higher, but not statistically significant (17 vs. 26%; p = .07). Mean

**Funding:** The authors received no specific funding for this work.

**Competing interests:** The authors have declared that no competing interests exist.

follow-up was 25.7 months, lower for ATOMS than ProACT (20.8 vs. 30.6 months; p = .02). The rate of working devices favoured ATOMS at 1-year (92 vs. 76; p < .0001), 2-years (85 vs. 61%; p = .0008) and 3-years (81 vs. 58%; p = .0001). Significant heterogeneity was evidenced, due to variable incontinence severity baseline, difficulties for a common reporting of complications, different number of adjustments and time of follow-up and absence of randomized studies.

## Conclusions

Despite the limitations that studies available are exclusively descriptive and the follow-up is limited, literature findings confirm ATOMS is more efficacious, with higher patient satisfaction and better durability than ProACT to treat male stress incontinence.

## Introduction

Urinary incontinence can severely impact the quality of life of prostate cancer survivors after radical prostatectomy and/or radiotherapy [1,2]. The artificial urinary sphincter (AUS) has been the main therapy after failed conservative treatment but reposition of the posterior urethra using retrobulbar slings has been a great revolution [3]. Also development of new devices usig postoperative adjustment has allowed a more personalized approach. Systematic reviews on the topic tend to annalyze a variety of devices with different modes of action [4–7].

Adjustable continence therapy (ProACT®, Uromedica, Minneapolis, MN) using balloons placed periurethraly near the bladder neck was the first adjustable implant to treat stress incontinence. It consists of two volume-adjustable balloons, that increase urethral resistance during voiding. Adjustable transobturator male system (ATOMS®, A.M.I., Feldkirch, Austria) is another adjustable device that compresses the bulbar urethra ventrally but, unlike ProACT, its two-arm mesh allows firm transobturator fixation on both sides of the ischio-pubic bone. Both devices can be adjusted postoperatively without need for aneasthesia. Besides, they open a new perspective of treatment for patients with cognitive impairment and/or limited dexterity, as no manipulation is needed for voiding [8]. Unfortunately, prospective controlled studies comparing anti-incontinence devices are lacking [9].

ATOMS and ProACT share two important peculiarities. Their mode of action is through urethral compression, and both are easily adjusted postoperatively by percutaneous injection of sterile saline solution until continence or a maximun filling volume is accomplished. Their main differences stand in that compression is ventral in ATOMS and lateral (both sides) in ProACT, and also that ATOMS is fixed to the pelvis while ProACT is a non-fixed device. ProACT has longer been used as its first description for male stress incontinence was in 2005 [10] and ATOMS started in 2011 [8]. ProACT remains unchanged but several modifications have been undertaken in ATOMS port location and design, evolving from inguinal to scrotal and silicone-covered.

Randomized comparative studies between ProACT or ATOMS have not been performed to date but a number of individual or multi-center studies with same or very similar endpoints covered are available for both devices. With this study we aim to evaluate the current evidence on the efficacy and safety ATOMS compared to ProACT using a systematic review approach and meta-analysis.

## Materials and methods

A systematic review of the scientific literature was carried out in September 2019. Preferred Reporting Items for Systematic Reviews and Meta-Analyses (PRISMA) checklist is included (**S1 Table**). The search included studies published between January 2005 and August 2019. The search was undertaken in PubMed, Embase, Web of Science and Scopus. Search strategy was designed according to PICOS criteria (Population, Intervention, Comparator, Outcomes and Study Design) (**Table 1**) for the identification of studies using free and controlled terminology. Search strategy included the terms: "Urinary incontinence" AND "male" AND "ATOMS" OR "ProACT". A manual revision of the bibliographic references was also carried out. We included prospective and retrospective, multi-centre or single-centre case series, published in English, Spanish, German or Italian on patients treated with ATOMS or ProACT devices for SUI after prostate surgery. No clinical trial on the topic could be found. Duplicate studies, editorial comments, letters to the editor or expert opinions in non-systematic reviews, a report after cystoprostatectomy and a meta-analysis centered only on ATOMS were excluded.

The identification, selection and extraction of data from the studies were carried out exhaustively and independently by two reviewers. A first selection of the studies was performed by reading the title and the abstract, and those that met the inclusion criteria were reviewed by reading the full text. Disagreements were resolved by consensus or in collaboration with another member of the research team. Consecutive publications from the same groups were evaluated to avoid overlapping of patients in different studies. Whenever repetition was detected we only used the last publication with larger number of patients. The evidence was summarized with the data extracted from the studies that included bibliographic information, characteristics of the study and patients, characteristics of the intervention, and outcome measures in relation to the efficacy and safety of the procedure. The definitions of postoperative dryness and improvement were comparable among studies. Differential pad-count or differential 24h pad-test was calculated as after adjustment minus baseline when available, assuming both values are non-dependent. Safety was defined according to rate of explantation and complications.

Primary efficacy indicator was percentage of dry patients after adjustment, defined as patients using no pad or only one safety pad/day (PPD). When dry rate change evolution in time was described in a study only the initial report was used. Main secondary efficacy outcome was percentage of overall improvement (defined as ≥50% decrease in pad-count and/or in 24h pad-test). Differential pad-count and differential 24h pad-test between baseline and after adjustment were investigated to evaluate the magnitude effect. The proportion of patients satisfied with the intervention, complication rate, explantation rate, number of adjustments, follow-up after implantation and durability of the devices were also evaluated as secondary

**Table 1. PICOS criteria to guide the systematic review.**

| | |
|---|---|
| **Population** | Males with mild, moderate or severe stress urinary incontinence after prostate surgery, eithr previously radiated or not and treated primarily or after failure of other surgical devices |
| **Intervention** | Placement of ATOMS ® device |
| **Comparison** | Placement of ProACT ® device |
| **Outcomes** | Primary: Overall dryness rate (no pad or one security pad per day)<br>Secondary: Overall improvement rate, differential pad-count and/or pad-test (after adjustment with respects to baseline), complication rate and durability |
| **Study Design** | Retrospective and prospective case series |

outcomes. We also included an analysis of manuscript quality assessment using the Newcastle-Ottawa scale.

The heterogeneity assumption was evaluated by the chi-square-based Cochran's Q test (which was considered significant at p-value < .05), and quantified with the I2 statistic (with values <25%, 25% to 75%, and >75% interpreted as representing low, moderate and high levels of heterogeneity, respectively), and with tau2 (the between-study variance) that was obtained with the DerSimonian and Laird method. Due to the presence of heterogeneity, random-effects model with the inverse variance method was used for pooling of single proportions, single means or mean differences from primary studies. The pooled effect was described with 95% confidence interval (CI). Publication bias was assessed by visual inspection of funnel plots and quantified by the Egger's linear regression test. Statistical analysis and figures were performed with the meta package of R software version 3.4.1 (R Foundation for Statistical Computing, Vienna, Austria) [11].

## Results

### Literature search

The search initially provided 174 references, 61 of which dealt with the topic evaluated. Additional records identified 6 further references through other sources. The reason for exclusion was multiple. Three studies were duplicated and 13 were not considered eligible (11 did not meet inclusion criteria and 2 were previous non-updated meta-analysis). Besides, 10 full-text articles were excluded for several reasons: 2 described the operative technique and urodynamic data but did not provide information to be included, 1 used different devices simultaneously, 1 was a case report with an unusual complication, 3 were pilot studies published before the definite article included and 3 were post-hoc studies of primary reports already included that focused on specific results (patient reported outcomes, stratification of results according to different generation of devices and a subgroup of patients with longer follow-up) (**Fig 1**).

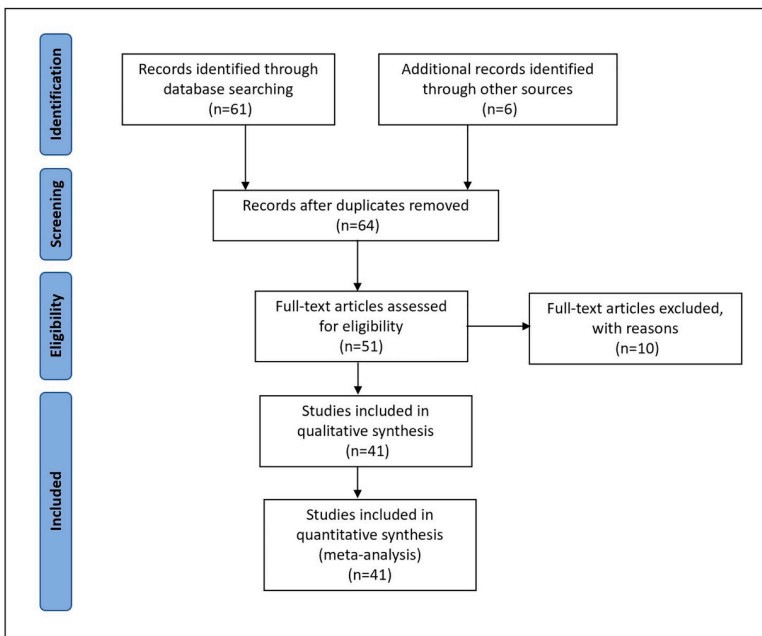

**Fig 1. Flow chart with studies included in the systematic review.**

## Characteristics of the studies

**Tables 2** and **3** show the characteristics of the 41 selected studies, published between 2005 and 2019, that include 3059 patients in total. Twenty-one articles were published using ProACT [12–32] and 20 using ATOMS [33–52]. No study was randomized controlled, and there was no comparison reported among these or other devices. The main etiology of incontinence was radical prostatectomy but other causes were included. Also the percentage of radiated patients and the degree of baseline incontinence was variable. Dryness rate and improvement rate were very homogeneously reported among the studies. The proportion of patients self-considering satisfied with the device was also often specified; however, 24-h pad-count and, more interestingly, pad-test (baseline, after adjustment and differential) was absent in some studies. The Freeman-Tukey arcsine transform was developed (**S1 Fig**) and pooled proportion (dryness, improvement, satisfaction, explant and complication rates) and mean (number of fillings, differential pad-count and differential pad-test) with 95% confidence intervals were used as summary effect measure.

**Tables 4** and **5** summarize the different domains to evaluate study quality. Observational cohorts available have been analyzed according to the Newcastle-Ottawa Scale for quality assessment. None of the studies included in this meta-analysis was a randomized clinical trial and studies generally lack a nonexposed cohort or control. However, the comparision of the intervention is possible with respects to baseline situation before use of device. Attrition bias seems low and the cohorts appear comparable for outcome assessment.

## Baseline characteristics

Similarities between the populations evaluated for each device were investigated to confirm the groups are comparable. Firstly, regarding baseline incontinence severity baseline pad-count for ATOMS and ProACT were equivalent (5.05 (95% CI 4.8;5.3) PPD compared to 4.6 (95% CI 3.9;5.3) PPD; p = .27). Secondly, the proportion of patients with previous surgery for incontinence was similar between devices (16.1% (95% CI 9.7;23.5) for ATOMS vs. 11% (95% CI 5.05;18.6) for ProACT; p = .29). Finally, patient age at the time of surgery was higher for ATOMS (70.8 (95% CI 69.9;71.6) years vs. 68.9 (95% CI 68.1;69.6); p = .0009), but this difference is not clinically significant.

## Efficacy

Regarding the primary outcome, mean dryness rate was higher with ATOMS than ProACT (68% (95% CI 62;73) compared to 55% (95% CI 47;63); p = .01) (**Fig 2**). Also regarding the main secondary objective of the study, percentage of overall improvement was also higher for ATOMS than proACT (91% (95% CI 87;94) vs. 80% (95% CI 72;87; p = .007) (**Fig 3**). A higher proportion of patients self-declared satisfied with ATOMS as well (87% (95% CI 81;92) vs 56% (95% CI 36;76); p = .002) (**Fig 4**). Conversely, mean number of fillings for adjustment was lower for ATOMS than proACT (2.4 (95% CI 1.9;2.9) vs. 3.5 (95% CI 3;3.9); p = .001) (**Fig 5**).

Results revealing the magnitude of effect for improving urine loss are quantitatively notorious. Mean baseline pad-count was equivalent in the studies evaluated (5 PPD (95% CI 4.8;5.3) for ATOMS and 4.6 PPD (95% CI 3.9;5.3) for proACT; p = .27) but postoperative pad-count was lower for ATOMS (1.1 PPD (95% CI .9;1.3) vs. 2.1 PPD (95% CI 1.7;2.5); p < .0001). Therefore, mean differential pad-count was also lower for ATOMS (-4 PPD (95% CI -4.4;-3.7) vs. -2.5 PPD (95% CI 3.5;-1.5); p = .005) (**Fig 6**). Studies revealing mean pad-test were less numerous, especially for proACT; however, interesting data were also evidenced. Mean baseline pad-test reported was significantly higher for ATOMS (505 mL (95% CI 461;549) vs. 312 mL (95% CI 231;394); p < .0001) but postoperative pad-test became equivalent (22 mL (95%

**Table 2. Studies with ProACT included in the meta-analysis and variables evaluated.**

| Author Year (Reference) | N | Dry rate (%) | Definition of dryness | Improved rate (%) | Number of adjustments | Satisfied rate (%) | Explant rate (%) | Complication rate (%) | Major complication rate (%) | Baseline pad count (PPD) | Postoperative pad count (PPD) | Baseline pad test (mL/day) | Postoperative pad test (mL/day) | Previous surgery for incontinence (%) | Mean age (yrs) | Mean follow-up (mo) |
|---|---|---|---|---|---|---|---|---|---|---|---|---|---|---|---|---|
| Hübner & Scharp 2005 (12) | 117 | 67 | 0-1 PPD | 88 | 3(1-15) | NR | 27.4 | 46.2 | NR | 5.6±3.8 | 2.5±2.5 | NR | NR | NR | 70 (50-89) | 13(3-54) |
| Trigo-Rocha et al 2006 (13) | 25 | 65.2 | 0-1 PPD | 78.2 | 4.6(1-7) | NR | 17.3 | NR | NR | 4.7±1.7 | 1.8±1.6 | NR | NR | NR | 68.6 (61-72) | 22.4(6-48) |
| Kocjancic et al 2007 (14) | 64 | 67 | 0-1 PPD | 82 | 3(0-9) | NR | 14 | 17.2 | NR | 5.2 | 3.6±3.3 | NR | NR | NR | 65.4 (25-79) | 19.5 (12-62) |
| Lebret et al 2008 (15) | 62 | 30 | 0 PPD | 89 | NR | NR | 30.6 | NR | NR | 4.6(1-10) | 1.1(0-6) | NR | NR | NR | 71.1 (52-87) | 6 |
| Gilling et al 2008 (16) | 37 | 62 | 0 PPD | 82 | 3.3(0-7) | NR | 13 | 48.6 | 16.2 | 2.8±2 | 1.6±1.5 | NR | NR | NR | 69.9 (59-79) | 51.5 (24-60) |
| Crivellaro et al 2008 (17) | 46 | 68 | 0-1 PPD | 85 | 3(1-7) | NR | 14 | NR | NR | 5.1 | 2.5 | NR | NR | NR | 67 (45-82) | 19 |
| Martens et al 2009 (18) | 29 | 31% | NR | NR | NR | 56 | 44.8 | 69 | 27.6 | 4.8 | 3.1 | NR | NR | NR | 65 | 41 |
| Gregori et al 2010 (19) | 79 | 66.1 | 0-1 PPD | 91.9 | 3.6(0-14) | NR | NR | 10.1 | NR | 3.5(1-10) | NR | 407.5 (40-1300) | NR | NR | 68 (51-82) | 25 |
| García-Matres et al 2009 (20) | 69 | 69.8 | 0-1 PPD | 84 | 3 | NR | NR | NR | NR | NR | NR | NR | NR | NR |  | 22(3-48) |
| Giammò et al 2010 (21) | 18 | NR | 0-1 PPD | 61 | 3(1-6) | 7 | NR | NR | NR | NR | NR | NR | NR | NR | NR | 24(12-38) |
| Rouprêt et al 2011 (22) | 128 | 66.4 | 0-1 PPD | 75 | 2.3(0-5) | NR | 18 | 25 | NR | 4.2(1-20) | 1.5 | NR | NR | 10 | 71 (52-87) | 56.3 (24-95) |
| Kjaer et al 2012 (23) | 114 | 50 | 0-1 PPD | 80 | 4(0-14) | 53 | 20.2 | 20.2 | NR | 4.75(1-26) | 2.25(0-26) | 352.5 (16-2800) | 11(0-3000) | NR | NR | 58(1-80) |
| Crivellaro et al 2012 (24) | 42 | 71 | 0-1 PPD | 92 | NR | 45 | 9.5 | NR | NR | NR | NR | NR | NR | 23.8 | 65.2 (21-80) | 12(3-19) |
| Gatti et al 2012 (25) | 28 | 60.7 | 0-1 PPD | 85.7 | NR | NR | 28.6 | 21 | NR | NR | NR | NR | NR | NR | NR | NR |
| Utomo et al 2013 (26) | 49 | 75.5 | 0-1 PPD | 83.7 | 4 | NR | 16.3 | NR | NR | NR | NR | NR | NR | 22.5 | NR | NR |
| Venturino et al 2015 (27) | 22 | 18 | 0 PPD | 82 | 4.5(0-15) | 45 | 55 | NR | NR | 5.9(3-12) | 3.9(0-12) | 242.3 (12-1200) | NR | NR | 70.2 (53-80) | 57 |
| Baron et al 2017 (28) | 14 | 57 | 0-1 PPD | 88 | NR | 77 | 28 | NR | NR | NR | NR | 95±130 | 34±83 | 100 | 69 (70-79) | 34(4-89) |

*(Continued)*

**Table 2.** (Continued)

| Author Year (Reference) | N | Dry rate (%) | Definition of dryness | Improved rate (%) | Number of adjustments | Satisfied rate (%) | Explant rate (%) | Complication rate (%) | Major complication rate (%) | Baseline pad count (PPD) | Postoperative pad count (PPD) | Baseline pad test (mL/day) | Postoperative pad test (mL/day) | Previous surgery for incontinence (%) | Mean age (yrs) | Mean follow-up (mo) |
|---|---|---|---|---|---|---|---|---|---|---|---|---|---|---|---|---|
| Nash et al 2018 (29) | 123 | 41 | 90–100% pad-weight reduction | 61 | NR | NR | 24.2 | 25.2 | NR | 4.1±2.3 | 2.8±1.8 | 399±437 | 216±322 | 22.5 | 69.7 ±7.9 | 18 |
| Nestler et al 2019 (30) | 134 | NR | NR | 82.6 | NR | NR | 52.7 | 8.2 | 2.2 | 6(4–7) | 1(1–2) | NR | NR | NR | 71 (67–75) | 26(9–59) |
| Noordhoff et al 2018 (31) | 143 | 47.4 | 0–1 PPD | 72.9 | 4(2–6) | 88.3 | 30.1 | 21.7 | NR | 3.5±3 | 1±1 | NR | NR | 14.7 | 69 (66–73) | 46(21–76) |
| Finazzi Agrò et al 2019 (32) | 240 | 29.6 | <8gr pad weight | 37.5 | 3.78(1–10) | 66.3 | 12.5 | 22.5 | 7.9 | NR | NR | 367±145 | 113±145 | 7.5 | 68.3 ±7.5 | NR |

PPD: pads per day; mL: mililiter; yrs: years; mo: months; NR: not reported

**Table 3. Studies with ATOMS included in the meta-analysis and variables evaluated.**

| Author Year (Reference) | N | Dry rate (%) | Definition of dryness | Improved rate (%) | Number of adjustments | Satisfied rate (%) | Explant rate (%) | Complication rate (%) | Major complication rate (%) | Baseline pad count (PPD) | Postopertive pad count (PPD) | Baseline pad test (mL/day) | Postoperative pad test (mL/day) | Previous surgery for incontinence (%) | Mean age (yrs) | Mean follow-up (mo) |
|---|---|---|---|---|---|---|---|---|---|---|---|---|---|---|---|---|
| Hoda et al 2012 (33) | 124 | 61.6 | 0–1 PPD | 93.8 | 4.3±1.8 | NR | 4 | 8.9 | NR | 8.8±3.8 | 1.8±1.2 | 725±372 | NR | NR | 71.2±5.5 | 19.2 ±2.2 |
| Seweryn et al 2012 (34) | 38 | 60.5 | 0–1 PPD <15mL | 84.2 | 3.97(0–9) | NR | 10.5 | NR | 15.8 | 6.78(2–10) | 1.36(0–10) | 747 (230–1600) | 115(0–1500) | 28.9 | 70(60–83) | 16.9 |
| Hoda et al 2013 (35) | 99 | 63 | 0–1 PPD <10mL | 92 | 3.8±1.3 | NR | 4 | NR | NR | 7.1±2.3 | 1.3±1.1 | 681±466 | 79.7±210 | 34.3 | 70.4±6.2 | 17.8 ±1.6 |
| Krause et al 2014 (36) | 36 | 39 | 0–1 PPD | 50 | NR | 61.8 | 30.5 | 44.4 | NR | 8.33 | 4.4 | NR | NR | 30.5 | 70.4(50–79) | NR |
| González-Pérez et al 2014 (37) | 13 | 92.3 | 0 PPD | 87.1 | NR | 100 | 0 | NR | NR | NR | NR | NR | NR | 15.4 | 63(59–87) | 16(4–32) |
| Friedl et al 2016 (38) | 34 | 56 | 0–1 PPD | 88 | NR | NR | 11.8 | 17.6 | NR | 3.5±0.2 | 1.5±0.3 | NR | NR | 29.4 | 70.7(55–83) | 5.7±0.5 |
| Mühlstädt et al 2016 (39) | 54 | 48.1 | 0 PPD | 77.7 | 4.5±2.3 | NR | 7.4 | 25.9 | NR | 7.7±4.8 | 1.6±1.7 | NR | NR | 20.4 | 67.5 ± 7.3 | 27.5 ±18.4 |
| Friedl et al 2016 (40) | 62 | 61.3 | 0–1 PPD | 90 | 1.5±1.2 | NR | 14.5 | 6.45 | NR | 4(3–5) | 1(0–2) | 350 (300–542) | 5(0–135) | 27.4 | 71.3(69–75) | 17.7 (1.7–55.5) |
| Hüsch et al 2016 (41) | 49 | NR | NR | NR | NR | NR | 2.0 | 14.2 | 2.0 | NR | NR | NR | NR | NR | NR | NR |
| Buresova et al 2017 (42) | 35 | 62.9 | 0–1 PPD | 100 | 4.3(1–15) | NR | 2.9 | 20 | NR | 5 | 1 | NR | NR | NR | 66.7(51–81) | 21.2(3–63) |
| Friedl et al 2017 (43) | 287 | 64 | 0–1 PPD <10mL | NR | 3(2–4) | NR | 19.5 | 7 | 2 | 4(3–5) | 1(0–2) | 400 (300–700) | 18(0–105) | NR | 70(66–74) | 31(10–54) |
| Friedl et al 2017 (44) | 49 | 57.1 | 0–1 PPD <10mL | 89.7 | 2±1 | NR | 19.5 | 34.7 | 16.3 | 4(3–6) | 1(0–2) | 458 (310–630) | 10(0–90) | 16.3 | 73 (68–76) | 32±8.5 |
| Angulo et al 2017 (45) | 34 | 85.3 | 0–1 PPD | 95 | 1±3 | 97 | 0 | 14.7 | NR | 5±3 | 0±0 | 510±500 | 0±15 | 11.8 | 70.5 (48–79) | 18.5 ±10 |
| Manso et al 2018 (46) | 25 | 64 | 0–1 PPD | 100 | 1.54±1.3 | 84 | 0 | NR | 4 | 4.84 ±2.95 | 1.6±2.02 | NR | NR | 12 | 71.4±6.6 | 21.56 ±8.89 |
| Angulo et al 2018 (47) | 215 | 80.5 | 0–1 PPD <10mL | 85.1 | 1.4±1.9 | 85.1 | 3.25 | 15.3 | 3.7 | 3.9±2 | 0.9±1.5 | 484±372 | 63.5±201 | 5.6 | 69.7±6.8 | 24.3 ±15 |
| Esquinas et al 2018 (48) | 60 | 81.7 | 0–1 PPD | 93.3 | 1±2 | 93.2 | 1.7 | 18.6 | NR | 5±3 | 0±1 | 465±450 | 0±20 | 6.7 | 72±7 | 21±22 |
| Angulo et al 2018 (49) | 34 | 75 | 0–1 PPD | 91.5 | 1±3 | 80 | 0 | 15 | NR | 4±3 | 0±1.5 | 375 ±855 | 10±32.25 | 10 | 76.5±9 | 38.5 ±19.5 |
| Angulo et al 2018 (50) | 25 | 64 | 0–1 PPD | 100 | 1±1 | 83.3 | 3.3 | 13.3 | NR | 4±3 | 0±1 | 435±393 | 10±30 | 100 | 73±10 | 21.56 ±8.89 |
| Giammò et al 2019 (51) | 52 | 73.1 | 0–1 PPD | 98.1 | 1.55±1 | NR | 0 | 19 | 0 | 4.23(2–8) | NR | 412 (180–1100) | 100(0–440) | 57.7 | 77.7(58–84) | 20.1 ±20.7 |

(*Continued*)

**Table 3.** (Continued)

| Author Year (Reference) | N | Dry rate (%) | Definition of dryness | Improved rate (%) | Number of adjustments | Satisfied rate (%) | Explant rate (%) | Complication rate (%) | Major complication rate (%) | Baseline pad count (PPD) | Postopertive pad count (PPD) | Baseline pad test (mL/day) | Postoperative pad test (mL/day) | Previous surgery for incontinence (%) | Mean age (yrs) | Mean follow-up (mo) |
|---|---|---|---|---|---|---|---|---|---|---|---|---|---|---|---|---|
| Doiron et al 2019 (52) | 160 | 80 | 0–1 PPD | 87.8 | 2.4 | 86.3 | NR | 22.3 | 4.4 | 4(3–5) | 0.5(0-1-9 | NR | NR | 16.3 | 70.5±6.6 | 9(4.5–13.5) |

PPD: pads per day; mL: milliliter; yrs: years; mo: months; NR: not reported

**Table 4. Newcastle-Ottawa scale for assessing the quality of cohort studies with ProACT included in the meta-analysis.**

| Author Year (Reference) | Selection | | | | Comparability (e) | Outcome | | |
|---|---|---|---|---|---|---|---|---|
| | Representativeness of the exposed (interventional cohort) (a) | Selection of the nonexposed cohort (b) | Ascertainment of exposure (intervention) (c) | Incident disease (d) | | Assessment of outcome (f) | Length of follow-up (g) | Adequacy of follow-up (h) |
| Hübner & Schlarp 2005 (12) | A | C | A | A | A | B | A | A |
| Trigo-Rocha et al 2006 (13) | A | C | A | A | A | B | A | B |
| Kocjancic et al 2007 (14) | A | C | A | A | A | B | A | A |
| Lebret et al 2008 (15) | A | C | A | A | A | B | B | A |
| Gilling et al 2008 (16) | A | C | A | A | A | B | A | A |
| Crivellaro et al 2008 (17) | A | C | A | A | A | B | A | A |
| Martens et al 2009 (18) | A | C | A | A | A | B | A | A |
| Gregori et al 2010 (19) | A | C | A | A | B | D | A | A |
| García-Matres et al 2009 (20) | A | C | C | A | B | D | A | D |
| Giammò et al 2010 (21) | A | C | D | A | B | D | A | D |
| Rouprêt et al 2011 (22) | A | C | A | A | A | B | A | A |
| Kjaer et al 2012 (23) | A | C | A | A | A | B | A | A |
| Crivellaro et al 2012 (24) | A | C | A | A | B | C | A | A |
| Gatti et al 2012 (25) | A | C | D | A | B | D | B | D |
| Utomo et al 2013 (26) | A | C | D | A | B | D | B | D |
| Venturino et al 2015 (27) | A | C | A | A | A | B | A | A |
| Baron et al 2017 (28) | C | C | A | A | A | B | A | A |
| Nash et al 2018 (29) | A | C | A | A | A | B | A | A |
| Nestler et al 2018 (30) | A | C | A | A | A | B | A | A |
| Noordhoff et al 2018 (31) | A | C | A | A | A | B | A | A |
| Finazzi Agrò et al 2019 (32) | A | C | A | A | A | B | A | A |

(a) A, truly representative of the average patient at risk for male stress incontinence; B, somewhat representative of the average patient at risk; C, selected group; D, no description.

(b) A, drawn from the same source as the intervention cohort (concurrent controls); B, drawn from a different source (historical controls); C, no description of the derivation of the nonexposed control

(c) A, secure record; B, structures review; C, written self-report; D, no description.

(d) Demonstration that outcome of interest was not present at the start of the study: A, yes; B, no.

(e) Comparability of cohorts on the basis of the design or analysis: A, study controls for the most important factor (conditioning regimen), B, study controls for any additional factor; C, not carried out or not reported.

(f) A, independent blind assessment; B, record linkage; C, self-report; D, no description.

(g) Was follow-up long enough for outcomes to occur? A, yes; B, no.

(h) A, complete follow-up (all subjects were accounted for); B,Subject lost to follow-up were unlikely to introduce bias because small numbers were lost (>90% had follow-up, or description was provided of those lost); C, follow-up rate <90%, and there was no description of those lost; D, no statement.

**Table 5. Newcastle-Ottawa scale for assessing the quality of cohort studies with ATOMS included in the meta-analysis.**

| Author Year (Reference) | Selection | | | | Comparability (e) | Outcome | | |
|---|---|---|---|---|---|---|---|---|
| | Representativeness of the exposed (interventional cohort) (a) | Selection of the nonexposed cohort (b) | Ascertainment of exposure (intervention) (c) | Incident disease (d) | | Assessment of outcome (f) | Length of follow-up (g) | Adequacy of follow-up (h) |
| Hoda et al 2012 (33) | A | C | A | A | A | B | A | A |
| Seweryn et al 2012 (34) | A | C | A | A | A | B | A | A |
| Hoda et al 2013 (35) | A | C | A | A | A | B | A | A |
| Krause et al 2014 (36) | B | C | A | A | A | C | A | D |
| González-Pérez et al 2014 (37) | B | C | A | A | B | C | A | A |
| Friedl et al 2016 (38) | A | C | A | A | A | B | A | A |
| Mühlstädt et al 2016 (39) | A | C | A | A | A | B | A | A |
| Friedl et al 2016 (40) | A | C | A | A | A | B | A | A |
| Hüsch et al 2016 (41) | A | A | A | A | C | B | B | D |
| Buresova et al 2017 (42) | A | C | A | A | A | B | A | A |
| Friedl et al 2017 (43) | A | C | A | A | A | B | A | A |
| Friedl et al 2017 (44) | C | C | B | A | B | B | A | A |
| Angulo et al 2017 (45) | A | C | A | A | A | B | A | A |
| Manso et al 2018 (46) | A | C | A | A | A | B | A | A |
| Angulo et al 2018 (47) | A | C | A | A | A | B | A | A |
| Esquinas et al 2018 (48) | A | C | A | A | A | B | A | A |
| Angulo et al 2018 (49) | C | C | A | A | A | B | A | A |
| Angulo et al 2018 (50) | C | C | A | A | A | B | A | A |
| Giammò et al 2019 (51) | B | C | A | A | A | B | A | A |
| Doiron et al 2019 (52) | A | C | A | A | A | B | A | A |

(a) A, truly representative of the average patient at risk for male stress incontinence; B, somewhat representative of the average patient at risk; C, selected group; D, no description.

(b) A, drawn from the same source as the intervention cohort (concurrent controls); B, drawn from a different source (historical controls); C, no description of the derivation of the nonexposed control

(c) A, secure record; B, structures review; C, written self-report; D, no description.

(d) Demonstration that outcome of interest was not present at the start of the study: A, yes; B, no.

(e) Comparability of cohorts on the basis of the design or analysis: A, study controls for the most important factor (conditioning regimen), B, study controls for any additional factor; C, not carried out or not reported.

(f) A, independent blind assessment; B, record linkage; C, self-report; D, no description.

(g) Was follow-up long enough for outcomes to occur? A, yes; B, no.

(h) A, complete follow-up (all subjects were accounted for); B,Subject lost to follow-up were unlikely to introduce bias because small numbers were lost (>90% had follow-up, or description was provided of those lost); C, follow-up rate <90%, and there was no description of those lost; D, no statement.

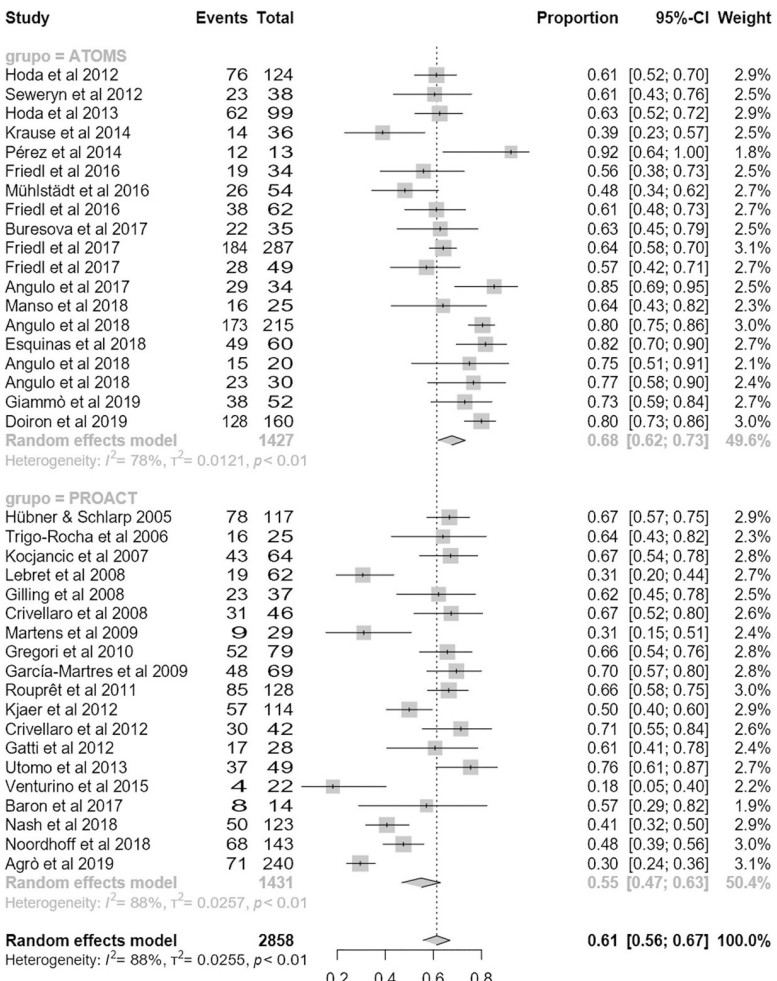

**Fig 2. Forest plot of studies analyzed for dry rate.**

CI 13;31) for ATOMS and 91 mL (95% CI 15;186) for ProACT; p < .08). Not strikingly, mean differential pad-test was lower for ATOMS (-426 mL (95% CI -462;-389) vs. -211 mL (95% CI -312;-111); p < .0001) (**Fig 7**).

Risk of publication bias was not identified for primary efficacy outcome, and variables evaluating the effect magnitude (differential pad-test and differential pad-count) (**S2 Table**). However, significant heterogeneity was detected in all variables evaluated (**S3 Table**), what possibly depends on different baseline severity of incontinence and patient profile, use of differrent generation devices for ATOMS and also different technique to appropriately place ProACT as described in the literature. Only randomization could reduce these biases.

## Safety and device durability

Mean follow-up was 25.7 months (95% CI 22.7;28.7), lower for ATOMS than ProACT (20.8 months (95% CI 17.1;24.5) vs. 30.6 months (95% CI 23;38.1); p = .02). The studies presented interesting safety results, both on explatation and complications. Mean explant rate was significantly lower for ATOMS (5% (95% CI 2;9) vs. 25% (95% CI 19;31); p < .0001) (**Fig 8**). Complication rate was also lower for ATOMS, but without reaching statistical significance (17% (95% CI 13;22) vs. 26% (95% CI 18;34); p = 0.067) (**Fig 9**). Similarly, major complication rate

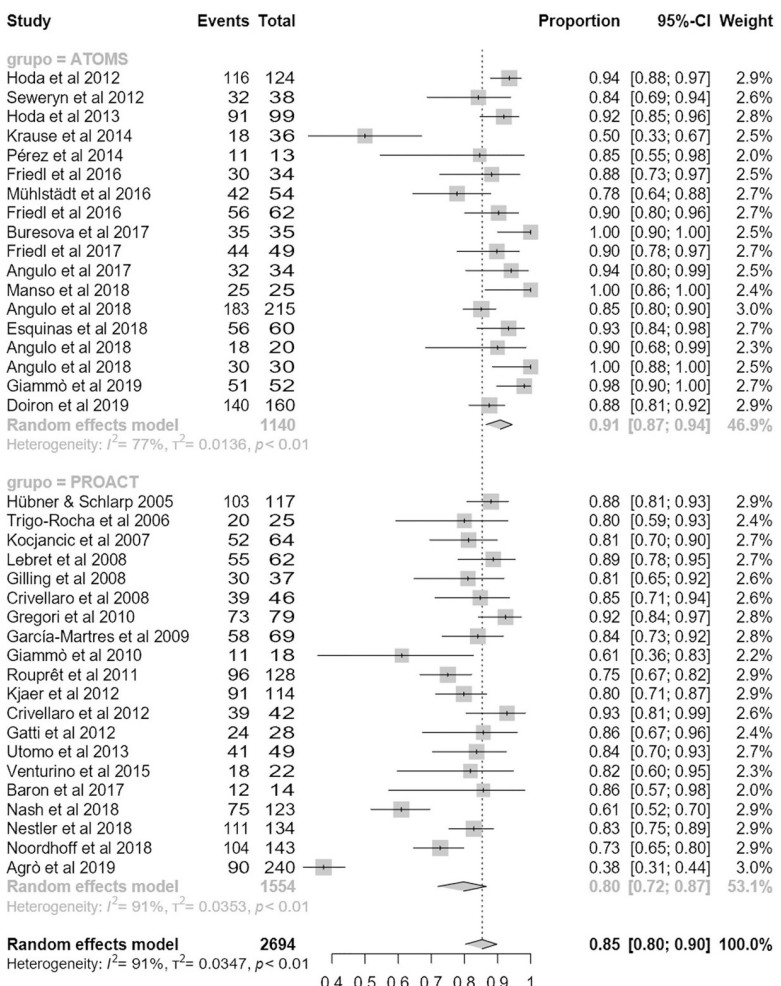

**Fig 3. Forest plot of studies analyzed for improve rate.**

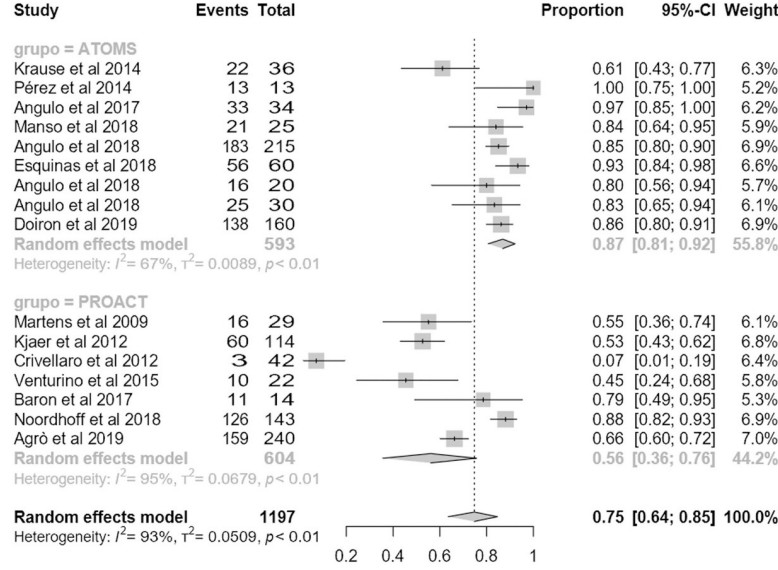

**Fig 4. Forest plot of studies analyzed for satisfaction rate.**

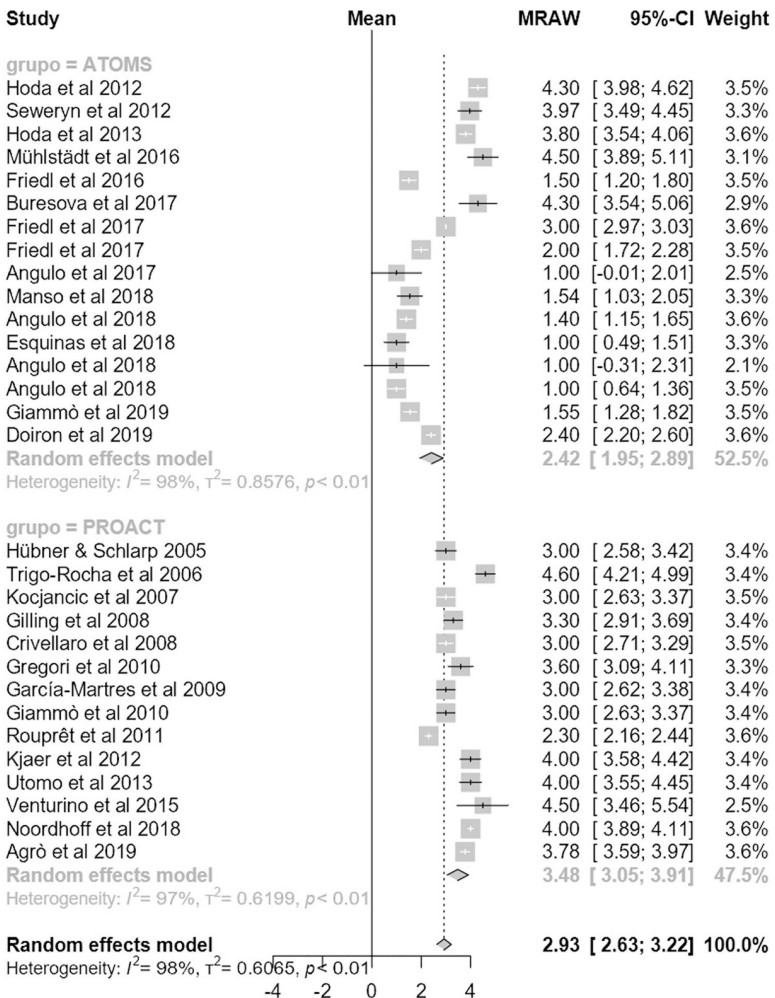

**Fig 5. Forest plot of studies analyzed with number of fillings.**

was similar between devices 4.2% (95% CI 1.7;7.7) for ATOMS vs. 10.4% (95% CI 3.15;20.7) for ProACT; p = 0.15). Risk of publication bias was not identified for these safety outcomes, but significant heterogeneity was probably related to the variability in reporting complications among studies (**S2 Table**, **S3 Table**),

Only 4 studies reveal strong data on durability (2 for each device) and specific analysis of Kaplan-Meier curves given and number of patients at risk allows to compare the proportio of devices that continue working in place, was higher for ATOMS on 1-year (92% (95% CI 87;96) vs. 76% (95% CI 69;83); p < .0001), 2-years (85% (95% CI 74;93) vs. 61% (95% CI 52;70); p = .0008) and 3-years (81% (95% CI 73;87) vs. 58% (95% CI 48;67); p < .0001) (**Fig 10**).

## Discussion

Despite male SUI therapy is an area of increasing interest the overall risk of bias must be reported as the majority of studies available about all male incontinence devices including AUS, slings and adjustable devices are retrospective or quasi prospective and mostly single center uncontrolled cohorts. In the absence of randomized controlled trials systematic reviews and meta-analysis can provide better evidence than simple prospective cohorts. Even though,

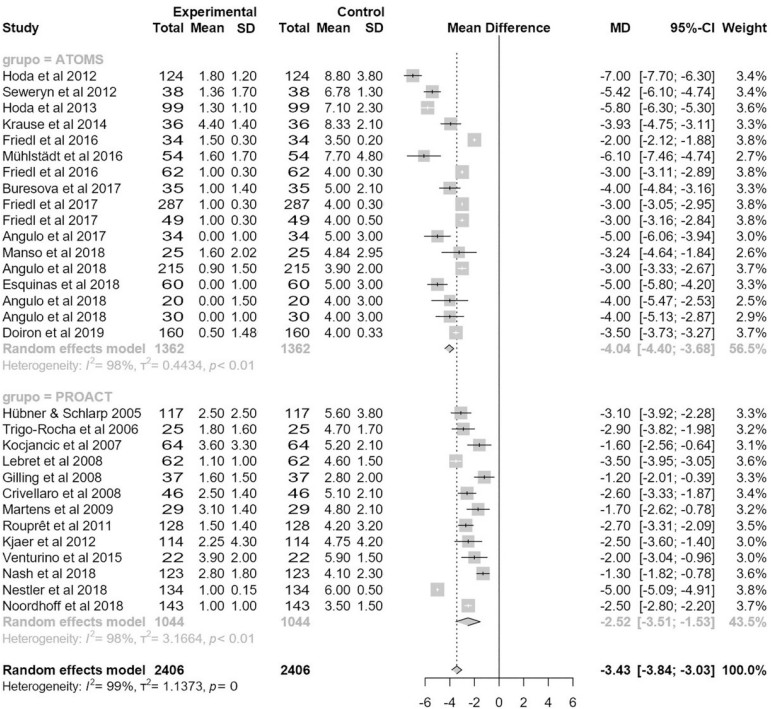

**Fig 6. Forest plot of studies analyzed with differential pad-count in pads-per-day.**

systematic reviews on the topic are scarce and confusing as tend to compile multiple devices with very different mode of action [4]. Also reviews are often biased to compare AUS and other devices, sometimes all named under the misleading term "slings". Adjustable slings developed to achieve urethral tensioning and results can be satisfactory even in severe incontinence and/or previous irradiation. However, current evidence is not firm to consider they are more efficacious than fixed male slings [6,7].

Both ProACT and ATOMS are really adjustable devices and unlike the AUS exert a non-circumferential periurethral compression that can be increased postoperatively in an office-

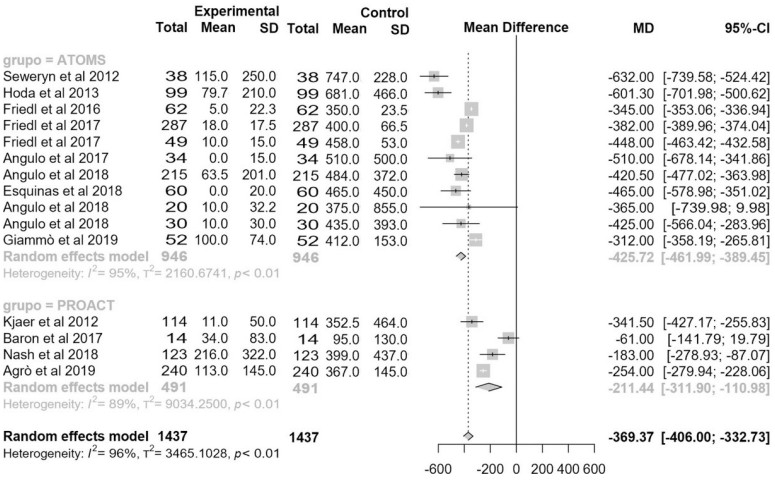

**Fig 7. Forest plot of studies analyzed with differential pad-test expressed in mL.**

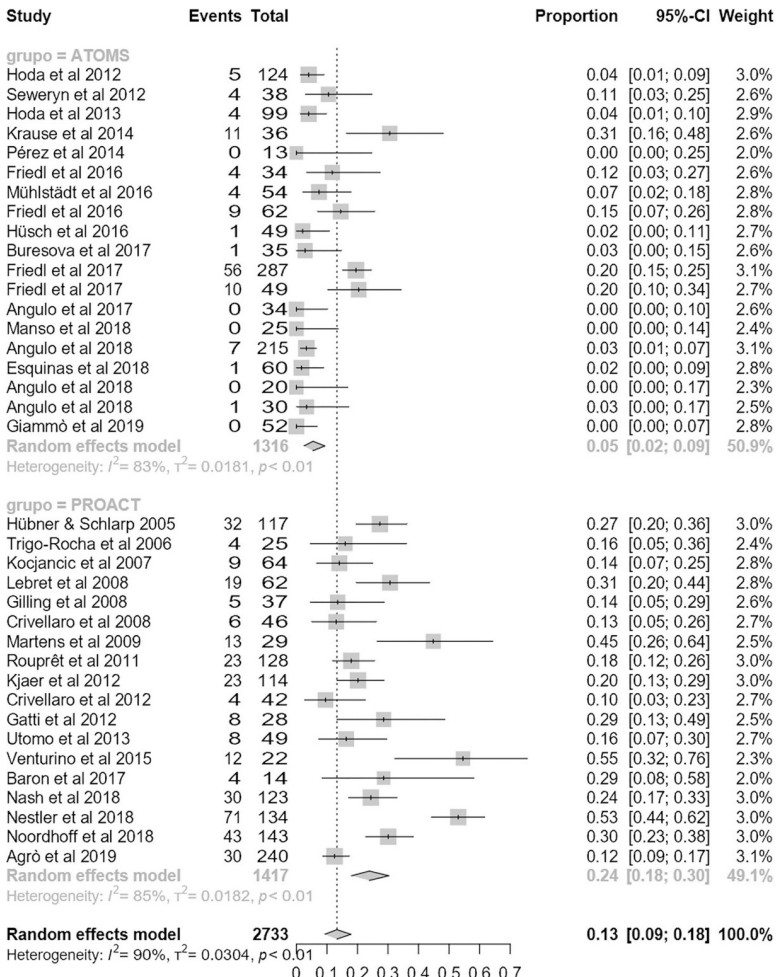

**Fig 8. Forest plot of studies analyzed for explant rate.**

based setting without re-intervention. The patient carries out urination without manipulation and the risk of mechanical failure is minimal, due to simplicity. Recent systematic reviews and meta-analysis confirm both ATOMS and ProACT are efficacious and safe alternatives to treat male stress incontinence of different degree but provide no comparison between them [53,54]. Interestingly, individual data reported for each device in those meta-analysis is very consistent with data here presented.

ATOMS has been implanted in Europe for a decade. It is used in mild and moderate-to-severe SUI, also in patients with previous radiotherapy. However, results are better in non-radiated patients and non-severe SUI [43,47,53]. Factors determinant of best patient perception with ATOMS are postoperative dryness, severe baseline incontinence severity, non-radiation, less pain at discharge and absence of postoperative complications [55]. ProACT has been implanted for two decades in Europe and more recently in the United States, often in outpatient basis under X-ray or transrectal ultrasound guidance. Based on this minimally invasiveness it has been recommended as first-line treatment in non-irradiated patients with mild-to-moderate incontinence [6,54,56], although revision and explantation rates are high [18,27,29,30] because a correct positioning of the balloons is mandatory to achieve good results [21]. Based on our findings, ProACT does not appear to be an ideal device for durable

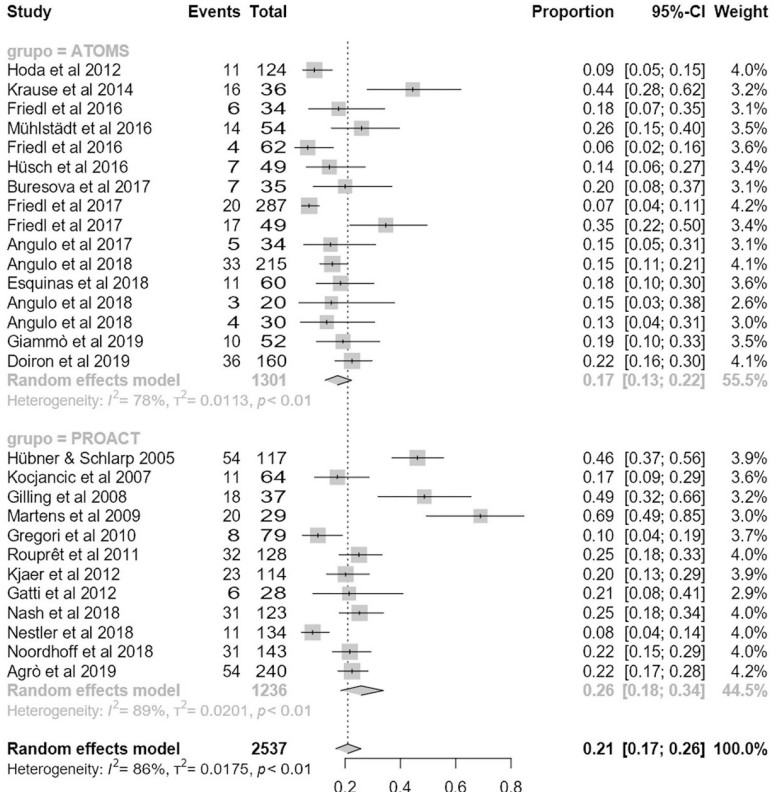

**Fig 9. Forest plot of studies analyzed for complication rate.**

continence or patient satisfaction. Also it is not so minimally invasive procedure as the complication rate, both total and major complications, surmounts that of ATOMS implant. A very recent experience uses ATOMS after retrieval of ProACT and indirectly supports ATOMS is superior to ProACT in the short-term [51]. What is more ATOMS can be used after radiation while ProACT is not recommended in this situation [6,31].

Pooled data in this meta-analysis reveals ATOMS is superior to ProACT in all the items evaluated for efficacy: dryness (68% vs 55%), improvement (91% vs 80%) and patient satisfaction (87% vs 56%). Definitions used as outcome measures are equivalent in most of the series analyzed (**Tables 2** and **3**). The magnitude of effect is also higher for ATOMS than ProACT, regarding both pad-count change (-4 vs -2.5 PPD) and pad-test change (-426 vs -211 mL). Regarding safety, ATOMS is again superior to ProACT with less explant rate (5% vs 24%) and a higher proportion of devices working in place during the first three years of follow-up.

The main limitations of this meta-analysis stand in the short follow-up, especially in the ATOMS arm, and in the very high heterogeneity observed between studies; probably reflecting a variable severity of sphincteric damage included and the absence of controlled randomized studies. Also the critria to report complications appear variable between the studies analyzed. The limitations highlighted are in consonance with the publication bias identified according to Egger's linear regression. Despite the existing limitations, we consider this meta-analysis can be of great help both for physicians and health care providers.

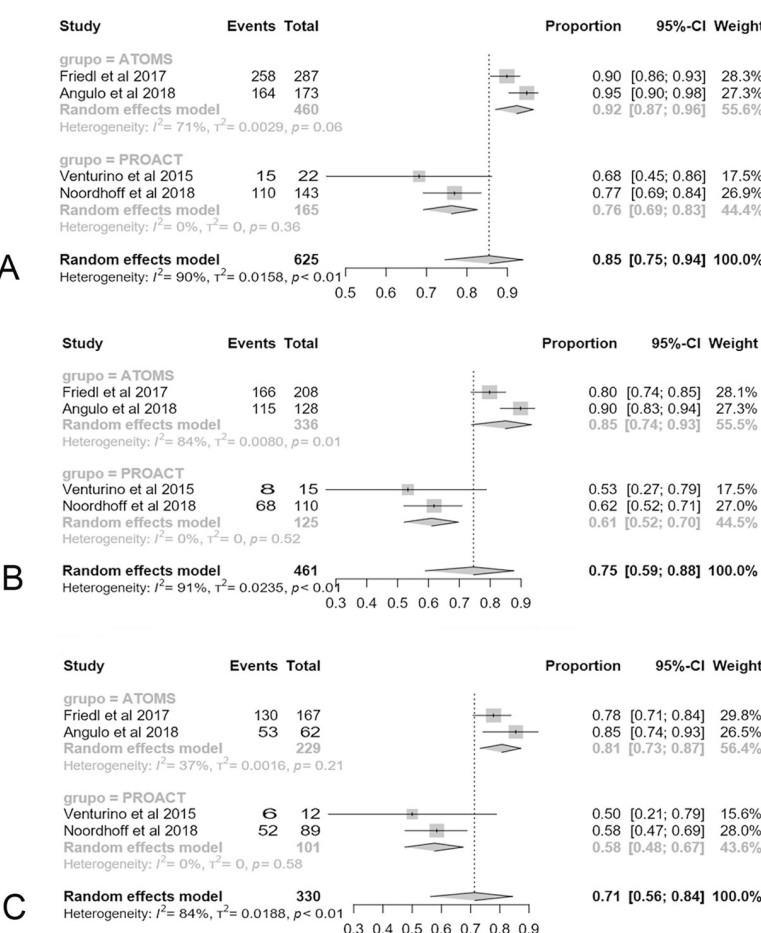

**Fig 10.** Forest plot of studies analyzed with durability of the devices at 12-months (A), 24-months (B) and 36-months (C).

## Conclusion

The decision upon which is the best device to treat male stress incontinence in a particular patient is influenced by patient status, manual dexterity, incontinence severity and previous radiotherapy. According to this systematic review and meta-analysis both the ProACT and the ATOMS system appear efficacious and safe procedures to treat male stress incontinence. However, taking into account the statistical summary of effect size ATOMS is a more efficacious alternative compared to ProACT with higher dryness, improvement and patient satisfaction rates, lower explant rate and higher durability. It must however be noted that the ATOMS studies have shorter follow-up than the ProACT studies.

## Supporting information

**S1 Table. Preferred Reporting Items for Systematic Reviews and Meta-Analyses (PRISMA) 2009 Checklist.**
(DOC)

**S2 Table. Assessment of the risk of publication bias by means of Egger's linear regression test for the study outcomes evaluated.**
(DOCX)

**S3 Table. Random effect model (proportion estimate, 95% Confidence Interval) and quantifying heterogeneity (I2, p-value) for the study outcomes evaluated.**
(DOCX)

**S1 Fig.** Freeman-Tukey double arcsine transform proportion for dryness (A), improvement (B), satisfaction (C), differential pad-count (D), differential pad-test (E), number of fillings (F), explant (G) and complication (H).
(TIF)

## Author Contributions

**Conceptualization:** Javier C. Angulo.

**Data curation:** Javier C. Angulo, Sandra Schönburg, Alessandro Giammò, Francisco J. Abellán, David Lora.

**Formal analysis:** Javier C. Angulo, Sandra Schönburg, Francisco J. Abellán, David Lora.

**Investigation:** Javier C. Angulo, David Lora.

**Methodology:** Javier C. Angulo, David Lora.

**Project administration:** Javier C. Angulo.

**Software:** Javier C. Angulo, David Lora.

**Supervision:** Javier C. Angulo, Sandra Schönburg.

**Validation:** David Lora.

**Visualization:** Javier C. Angulo, Alessandro Giammò, Francisco J. Abellán, Ignacio Arance, David Lora.

**Writing – original draft:** Javier C. Angulo, Sandra Schönburg, David Lora.

**Writing – review & editing:** Javier C. Angulo, Sandra Schönburg, Alessandro Giammò, Francisco J. Abellán, Ignacio Arance, David Lora.

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
