## [Decision Letter · Decision Letter 0]

11 Oct 2019

PONE-D-19-26165

Systematic review and meta-analysis comparing Adjustable Transobturator Male System (ATOMS) and Adjustable Continence Therapy (ProACT) for male stress incontinence.

PLOS ONE

Dear Dr Angulo,

Thank you for submitting your manuscript to PLOS ONE. After careful consideration, we feel that it has merit but does not fully meet PLOS ONE’s publication criteria as it currently stands. Therefore, we invite you to submit a revised version of the manuscript that addresses the points raised during the review process.

We would appreciate receiving your revised manuscript by Nov 25 2019 11:59PM. To enhance the reproducibility of your results, we recommend that if applicable you deposit your laboratory protocols in protocols.io, where a protocol can be assigned its own identifier (DOI) such that it can be cited independently in the future. For instructions see: http://journals.plos.org/plosone/s/submission-guidelines#loc-laboratory-protocols

We look forward to receiving your revised manuscript.

Kind regards,

Peter F.W.M. Rosier, M.D. PhD

Academic Editor

PLOS ONE

Journal Requirements:

Additional Editor Comments (if provided):

Dear authors,

In addition to the reviewers comments: Can you add an explanation for 'previous surgery' in the table. The search strategy as is reported in the manuscripts core tekst is fairly specific: have you missed publications because you seached brand names only? and or only "urinary incontinence" (word combination within quotation marks?) and or, preferably: add the search strategy to the supplements. Furthermore have you checked/controlled or corrected (N patients reported) for the fact that some publications may (partially) contain the same patient cohorts? And also; can you limit your discussion and conclusion more to what you found and concluded in your analysis. Comments about future research should be placed in the discussion and the discussion should not include e.g. 'technical' differences; this, as example, has not been the topic of your research. The sentence (discussion) '...current evidence is still low' should be made more specific; the (discussion and) conclusion could/should include a statement about the overall risk of bias in the publications (e.g. the proportion of retrospective or quasi prospective single center uncontrolled cohorts).

Reviewers' comments:

Reviewer's Responses to Questions

**Comments to the Author**

1. Is the manuscript technically sound, and do the data support the conclusions?

Reviewer #1: Yes

Reviewer #2: Yes

2. Has the statistical analysis been performed appropriately and rigorously? 

Reviewer #1: Yes

Reviewer #2: Yes

3. Have the authors made all data underlying the findings in their manuscript fully available?

Reviewer #1: Yes

Reviewer #2: Yes

4. Is the manuscript presented in an intelligible fashion and written in standard English?

Reviewer #1: Yes

Reviewer #2: Yes

5. Review Comments to the Author

Reviewer #1: Summary:

This systematic review compares two techniques for non-neurogenic male stress incontinence that can be adjusted post-operatively: ATOMS and ProACT. Efficacy and safety aspects were considered. The study was well performed and the manuscript is well written.

The conclusion was that ATOMS has better efficacy and a lower complication and explantation rate. It must however be noted that the ATOMS studies had a shorter follow-up than the ProACT studies (20.8 vs 30.6 months on average) and that the severity of complications was not described, only the complication rate. ProACT placement is a less invasive procedure than ATOMS placement. This is summarily hinted in the Discussion, but not described quantitatively. Nevertheless, the comparison of ProACT and ATOMS is fairly described.

Comments:

1. There are some typing errors throughout the manuscript, e.g., grupo instead of group. Also, sometimes a word is missing.

2. Results. The authors write: The main etiology of incontinence was prostatectomy but other causes were included. I assume they mean radical prostatectomy (RP) and not Millin or TURP. Can the authors report the percentages of men with RP in both the ATOMS and ProACT group? Are they comparable?

3. The authors write NA in Table 2. This probably means Not Applicable, but I think NR (Not Reported) would be better.

Reviewer #2: The authors present an interesting and important paper on male postprostatectomy incontinence. Two widely used devices are compared in this meta analysis and the results are presented clearely. Since there is only consensus, that the artifical sphincter should be considered gold standard, it ist important to evaluate the other existing devices. Especially since the artifical sphincter is not the ideal solution for every patient.

Minor spelling mistakes in the paper can easily be fixed. Nevertheless, one comment must be allowed. It would be useful and interesting, if not only the putcome parameters, but also patients characteristics would be compared. All studies have provided statistics concerning their patients like age and type of prior operation. Here the authors should show, that there are no significant differences between ProAct and Atoms groups, since this would make the conclusion stronger.

6. PLOS authors have the option to publish the peer review history of their article (what does this mean?). If published, this will include your full peer review and any attached files.

Reviewer #1: Yes: Jan Groen

Reviewer #2: Yes: Sebastian Nestler, UroGate

---

## [Author Response · Author response to Decision Letter 0]

7 Nov 2019

PONE-D-19-26165

Systematic review and meta-analysis comparing Adjustable Transobturator Male System (ATOMS) and Adjustable Continence Therapy (ProACT) for male stress incontinence.

Dear Editors,

First we would like to thank reviewers and academic editors of PLOS ONE for their effort and the recognition of our work. We are very happy to resubmit our improved work after considering all the recommendations given. 

This letter ('Response to Reviewers') is uploaded as separate file, together with a marked-up copy of the manuscript highlighting changes made ('Revised Manuscript with Track Changes'). 

The manuscript meets PLOS ONE's style requirements.

If the manuscript is finally accepted, I would like to make the peer review history publicly available.

RESPONSE TO EDITOR COMMENTS

1. As recommended the protocol has been deposited in protocols.io, and the DOI identifier assigned is: dx.doi.org/10.17504/protocols.io.8x9hxr6

2. As solicited by the Editor the label 'previous surgery' in the table is better specified as proportion of patients with previous surgery for incontinence.

3. The search strategy is specified in lines 102-103 in Material and Methods section. We confirmed we did not miss publications by using brand names or the combination of words "urinary incontinence".

4. We controlled consecutive publications from the same groups to avoid overlapping of patients in different studies. Whenever repetition was detected we only used the last publication with larger number of patients. This concept is important and is now stated in Material and Methods section (lines 115-118).

5. As recommended the discussion and conclusion sections have been limited more to what we found in your analysis. The sentence on comments about future research in the Conclusion section has been omitted. Similarly, in the Discussion section the sentences on the 'technical' differences between the devices has been omitted. 

6. The sentence in the Discussion '...current evidence is still low' is made more specific and instead the overall risk of bias is stated as all the studies available are retrospective or quasi prospective single center uncontrolled cohorts (lines 272-274).

7. The editor noticed some minor occurrence(s) of overlapping text with the following previous publication of our group. A meta-analysis based of the outcomes of ATOMS alone, published in Advances and Therapy (https://doi.org/10.1007/s12325-018-0852-4), reference 53. This and another meta-analysis on ProACT (reference 54) have been cited and discussed form the initial writing of this new article. We also paid great attention to avoid any duplicated text outside the Methods section.

8. As requested, we have included additional details in Methods section regarding an analysis of manuscript quality assessment using the Newcastle-Ottawa scale, lines 179-185 and new Table 3.

9. While revising the submission, figure files have been uploaded to the Preflight Analysis and Conversion Engine (PACE) digital diagnostic tool to ensure that figures meet PLOS requirements. 

RESPONSE TO REVIEWERS

Reviewer #1: 

Comment # 1. The conclusion was that ATOMS has better efficacy and a lower complication and explantation rate. It must however be noted that the ATOMS studies had a shorter follow-up than the ProACT studies (20.8 vs 30.6 months on average).

Response to Comment #1: This important detail is completed in the conclusion section (lines 330-332).

Comment # 2: The severity of complications was not described, only the complication rate. ProACT placement is a less invasive procedure than ATOMS placement. This is summarily hinted in the Discussion, but not described quantitatively. 

Response to Comment #2: The reviewer is right to state that this subject has not been properly addressed in the original version. In this revised version not only complications rate is described, but we have also reviewed and compared the proportion of major complications reported. That is lower in ATOMS than in ProACT, but again this difference did not reach statistical significance. Thereof we cannot conclude that ProACT is a less invasive procedure than ATOMS as described and this new idea is included in the revised version of the manuscript (lines 256-257, lines 302-303). 

Comment # 3. There are some typing errors throughout the manuscript.

Response to Comment #3: Some typing errors have been corrected. 

Comment # 4. Results. The authors write: The main etiology of incontinence was prostatectomy but other causes were included. I assume they mean radical prostatectomy (RP) and not Millin or TURP. Can the authors report the percentages of men with RP in both the ATOMS and ProACT group? Are they comparable?

Response to Comment #4: Thank you very much for this comment. The reviewer is right as we mean radical prostatectomy when we say prostatectomy. We have corrected that in the manuscript. It would be very interesting to compare the efficacy of devices not only in radical prostatectomy, but also un simple prostatectomy (Millin procedure) or transurethral resection of the prostate (TUR-P). However, that is not possible because most of the articles do not properly mention the proportion of cases derived from one or the other intervention, especially in the ProACT cohorts.

Comment # 5. The authors write NA in Table 2. This probably means Not Applicable, but I think NR (Not Reported) would be better.

Response to Comment # 5. NA is changed by NR (Not Reported) in Table 2.

Reviewer #2: 

Comment #1. Minor spelling mistakes in the paper can easily be fixed. 

Response to Comment #1: Some spelling mistakes ad minor typing errors have been corrected.

Comment # 2. Not only the outcome parameters, but also patient characteristics would be compared. All studies have provided statistics concerning their patients like age and type of prior operation. Here the authors should show, that there are no significant differences between ProAct and Atoms groups, since this would make the conclusion stronger.

Response to Comment #2: Thank you very much for this comment. The reviewer is right to ask not only to compare outcomes of the different devices, but also similarity between characteristics of the patients treated by one or the other device. Baseline pad-count (the most frequent objective measurement in most publications) is equivalent between cohorts. Patient age is included in new Table 2 for each study. Type of previous incontinence operation is not available for each cohort in the publications, as a variety of options are possible; however, we have analysed the proportion of patients with previous anti-incontinence operation. New pooled analysis of patient age at the time of implant, baseline pad-count and the proportion of patients previously intervened with both techniques are similar in the literature, what confirms cohorts are apt to be compared. As suggested by the reviewer these data are included both in Results section (lines 205-213) and in revised Supplementary material S3 and S4.

---

## [Decision Letter · Decision Letter 1]

13 Nov 2019

Systematic review and meta-analysis comparing Adjustable Transobturator Male System (ATOMS) and Adjustable Continence Therapy (ProACT) for male stress incontinence.

PONE-D-19-26165R1

Dear Dr. Angulo,

We are pleased to inform you that your manuscript has been judged scientifically suitable for publication and will be formally accepted for publication once it complies with all outstanding technical requirements.

With kind regards,

Peter F.W.M. Rosier, M.D. PhD

Academic Editor

PLOS ONE

Additional Editor Comments (optional):

Maybe you can follow the suggestion of reviewer 1; Q6, to mention the follow up duration in the abstract, I would be in favour of that.

Reviewers' comments:

Reviewer's Responses to Questions

**Comments to the Author**

1. If the authors have adequately addressed your comments raised in a previous round of review and you feel that this manuscript is now acceptable for publication, you may indicate that here to bypass the “Comments to the Author” section, enter your conflict of interest statement in the “Confidential to Editor” section, and submit your "Accept" recommendation.

Reviewer #1: All comments have been addressed

Reviewer #2: All comments have been addressed

2. Is the manuscript technically sound, and do the data support the conclusions?

Reviewer #1: Yes

Reviewer #2: Yes

3. Has the statistical analysis been performed appropriately and rigorously? 

Reviewer #1: Yes

Reviewer #2: Yes

4. Have the authors made all data underlying the findings in their manuscript fully available?

Reviewer #1: Yes

Reviewer #2: Yes

5. Is the manuscript presented in an intelligible fashion and written in standard English?

Reviewer #1: Yes

Reviewer #2: Yes

6. Review Comments to the Author

Reviewer #1: It would have been my preference to mention the mean follow-up durations not only in the article itself, but also in the abstract.

Reviewer #2: All comments have been adressed, I have no further questions or comments. The paper should be published

7. PLOS authors have the option to publish the peer review history of their article (what does this mean?). If published, this will include your full peer review and any attached files.

Reviewer #1: Yes: Jan Groen

Reviewer #2: No

---

## [Editor Report · Acceptance letter]

21 Nov 2019

PONE-D-19-26165R1 

Systematic review and meta-analysis comparing Adjustable Transobturator Male System (ATOMS) and Adjustable Continence Therapy (ProACT) for male stress incontinence. 

Dear Dr. Angulo:

I am pleased to inform you that your manuscript has been deemed suitable for publication in PLOS ONE. Congratulations! Your manuscript is now with our production department. 

With kind regards,

on behalf of

Dr. Peter F.W.M. Rosier 

Academic Editor

PLOS ONE